# Brain Invasion along Perivascular Spaces by Glioma Cells: Relationship with Blood–Brain Barrier

**DOI:** 10.3390/cancers12010018

**Published:** 2019-12-19

**Authors:** Simone Pacioni, Quintino Giorgio D’Alessandris, Mariachiara Buccarelli, Alessandra Boe, Maurizio Martini, Luigi Maria Larocca, Giulia Bolasco, Lucia Ricci-Vitiani, Maria Laura Falchetti, Roberto Pallini

**Affiliations:** 1Institute of Neurosurgery, Fondazione Policlinico Universitario A. Gemelli IRCCS, Università Cattolica del Sacro Cuore, 00168 Rome, Italy; s.pacioni@tiscali.it (S.P.); giorgiodal@hotmail.it (Q.G.D.); 2CNR-IBBC, Institute of Biochemistry and Cell Biology, 00015 Rome, Italy; marialaura.falchetti@cnr.it; 3Department of Oncology and Molecular Medicine, Istituto Superiore di Sanità, 0161 Rome, Italy; mariachiara.buccarelli@iss.it (M.B.); lriccivitiani@yahoo.it (L.R.-V.); 4Core Facilities, Istituto Superiore di Sanità, 00161 Rome, Italy; alessandra.boe@iss.it; 5Institute of Human Pathology, Fondazione Policlinico Universitario A. Gemelli IRCCS, Università Cattolica del Sacro Cuore, 00168 Rome, Italy; Maurizio.Martini@unicatt.it (M.M.); luigimaria.larocca@unicatt.it (L.M.L.); 6Epigenetics & Neurobiology Unit, European Molecular Biology Laboratory (EMBL Rome), 00015 Monterotondo, Italy; giulia.bolasco@embl.it

**Keywords:** blood–brain barrier, perivascular invasion, glioblastoma, glioma stem-like cells, brain endothelium

## Abstract

The question whether perivascular glioma cells invading the brain far from the tumor bulk may disrupt the blood–brain barrier (BBB) represents a crucial issue because under this condition tumor cells would be no more protected from the reach of chemotherapeutic drugs. A recent in vivo study that used human xenolines, demonstrated that single glioma cells migrating away from the tumor bulk are sufficient to breach the BBB. Here, we used brain xenografts of patient-derived glioma stem-like cells (GSCs) to show by immunostaining that in spite of massive perivascular invasion, BBB integrity was preserved in the majority of vessels located outside the tumor bulk. Interestingly, the tumor cells that invaded the brain for the longest distances traveled along vessels with retained BBB integrity. In surgical specimens of malignant glioma, the area of brain invasion showed several vessels with preserved BBB that were surrounded by tumor cells. On transmission electron microscopy, the cell inter-junctions and basal lamina of the brain endothelium were preserved even in conditions in which the tumor cells lay adjacently to blood vessels. In conclusion, BBB integrity associates with extensive perivascular invasion of glioma cells.

## 1. Introduction

Malignant gliomas are highly invasive cancers. The sub-cortical white matter and inter-hemispheric tracts, like the corona radiata and corpus callosum, are major paths for tumor spreading. However, glioma cells are also known to interact with the blood vessels, mainly in the way of vessel co-option, a phenomenon whereby the tumor cells organize themselves into cuffs around normal vessels [1]. In vivo studies showed that the vast majority of tumor cells located outside of the tumor bulk are in close relationship with the blood vessels, suggesting perivascular invasion [2,3,4]. Furthermore, a recent study that used clinically relevant xenograft models demonstrated that human glioma cells are able to migrate far away from the main tumor mass travelling between the endothelial cells and the endfeet of astrocytes, where even single glioma cells are sufficient to cause a focal breach of the blood–brain barrier (BBB) [5]. These results may carry important clinical implications and may refuel aggressive chemotherapy for treating brain areas of tumor infiltration. However, clinical experience suggests caution in translating these data directly to patients. For example, the brain surrounding glioblastoma (GBM) or anaplastic glioma, where histology reveals invasion by tumor cells, does not enhance on magnetic resonance (MR) after intravenous infusion of the contrast medium gadolinium (Gd), suggesting a preserved BBB. Then, one main objective of this study is to analyze the interaction of glioma cells with the brain vasculature. To address this issue, we used orthotopic xenografts of patient-derived glioma stem-like cells (GSCs) and surgical specimens. More specifically, we questioned whether the invading perivascular glioma cells do actually disrupt the BBB away from the tumor bulk, a condition whereby these cells would be no more protected from the reach of chemotherapeutic drugs. 

## 2. Results

### 2.1. Association of Glioma Cells with Blood Vessels and Disruption of BBB in Models of Brain Xenograft 

We first investigated in in vivo models the extent to which glioma cells associate with blood vessels in brain regions outside of the main tumor bulk and the relationships between perivascular invasion and BBB. Orthotopic xenografts were established in athymic rats using either the U87MG GBM cell line (*n*, 6), which shows poor ability to invade the brain and thus does not reflect the clinical situation, or the patient-derived GSC1 and GSC275 cell lines (*n*, 9), which develop highly infiltrating brain tumors in vivo similarly to what is seen in patients [6,7,8]. The GSC1 cell line had been established from a GBM of the proneural subtype and was molecularly characterized as a Glioma Stem full (GSf), a genotype that closely resemble the proneural one [9]. The GSC275 cell line had been raised from a GBM of the mesenchymal subtype and was molecularly characterized as a Glioma Stem restricted (GSr), resembling the mesenchymal subtype [9].

To visualize the pattern of brain invasion, the tumor cells were transduced to express GFP. Brain vasculature was stained with biotinylated Lectin *Lycopersicon esculentum* [10], a specific marker of endothelial cells. To assess the BBB, we used antibodies against the rat BBB (clone SMI-71), glucose transporter-1 (Glut-1), and zonula occludens (ZO)-1 protein (Appendix A). SMI-71 selectively stains the rat endothelial barrier antigen (EBA). This antigen is localized at the luminal side of brain endothelial cells [11] and its expression is highly decreased or even lost in areas of reduced BBB integrity [12]. Glut-1, a major glucose transporter across the mammalian BBB, is widely recognized as a specific marker of brain endothelium [13,14]. ZO-1 protein [15] is a key component of tight junctions (TJs) between adjacent endothelial cells, which primarily determine BBB permeability [16,17,18,19]. Alteration of ZO-1 expression causes TJ disorganization and leads to BBB disruption [5,20,21]. To detect vascular permeability, sections were stained with anti-rat IgG that highlights extravasated mouse immunoglobulins [22]. In brain xenografts, extravasation of these immunoglobulins correlates with vascular permeability, as assessed with Gd-enhanced MR [23]. 

Using these methods, we found that the U87MG xenografts elicited a strong neo-angiogenesis in the brain within 400 microns from the outer edge of the tumor (Appendix A). In this region, the newly formed vessels showed highly disrupted BBB, as demonstrated by the nearly absent SMI-71 staining and low ZO-1 expression (Appendix A). Only a few U87MG cells were able to invade the brain crossing the tumor-brain interface. Interestingly, these cells were nearly always associated with blood vessels showing some degree of BBB preservation (Appendix A). As expected, peritumor regions with reduced expression of SMI-71 and ZO-1 showed an intense anti-IgG staining, suggesting extravasation (Appendix A). 

Differently from the U87MG cells, GSC1 cells developed highly infiltrating brain xenografts. Tumor cells invaded the homolateral striatum and piriform cortex and extended contralaterally through the corpus callosum, anterior commissure, and septal nuclei. Analysis of the brain–tumor interface showed a great amount of cells invading into the brain using the white matter and blood vessels as scaffolds (Figure 1A). In the brain surrounding the xenograft, the vast majority of GSCs were associated with blood vessels in contact with the vascular surface (Figure 1B,C). GSCs laid outside the endothelial covering wrapping themselves around the abluminal surface or even fully encasing the blood vessels. Notably, such massive perivascular spreading of GSCs outside the main tumor mass occurred mainly along vessels with preserved BBB (Figure 1B,C and Appendix A). In particular, the SMI-71 reaction, which lacked almost completely in U87MG xenograft, was preserved in the vessels outside the tumor bulk of GSC1 xenografts. An inverse relationship was found between the density of tumor cells and SMI-71 staining, whereby in the tumor core, where tumor cell density was the highest, the vasculature expressed SMI-71 at very low levels (Figure 1D,E). Interestingly, GSCs laid around vessels with preserved BBB even at long distances from the tumor bulk. For example, in the caudate-putamen contralateral to the grafting site about 80 percent of vessels showing perivascular tumor infiltration had preserved BBB (Figure 1F,G). The BBB was preserved even in those vessels surrounded by multilayered tumor cells, as demonstrated by SMI-71 and ZO-1 staining (Figure 1H,I). In GSC275 brain xenografts, we found perivascular tumor cells spreading at distant sites from the bulk of the tumor (Appendix A). Importantly, even in the brain xenografts of the GSr subtype or mesenchymal-like cells, the BBB of vessels surrounded by tumor cells was not disrupted. 

### 2.2. Perivascular Invasion and Disruption of BBB in Human Specimens of Gliomas 

While the U87MG xenograft model confirmed that glioma cells invading along perivascular spaces breach the BBB outside the tumor bulk [5], brain xenografts of patient-derived GSCs gave quite different results. Then, we aimed at investigating the relationship between perivascular spreading and BBB disruption in tumor specimens from 21 patients suffering from glioma (Appendix A). Differently from in vivo models, where fluorescently labelled tumor cells can be easily traced through the brain, in human specimens the tumor cells are much more difficult to stain with specific markers [8]. Here, we used the antibody targeting Collapsin Response Mediator Protein 5 (CRMP5) that was recently proposed as a selective tumor marker for glioma cells [24,25]. Previously, we validated this antibody on patient-derived GSCs with mutant *IDH1* [26]. Here, we used surgical specimens of low-grade glioma (WHO grade II astrocytoma and oligodendroglioma) with mutant *IDH1/2* and showed a co-staining of the tumor cells with anti-IDH1 and anti-CRMP5 antibodies (Appendix A).

Tumor regions with variable degree of BBB permeability and vascularity on Gd-enhanced MR were located during surgery by navigational systems (Figure 2A,B). On Gd-enhanced MR, BBB disruption is evidenced by diffusion of IV-administered Gd, which diffuses from the blood to the brain interstitium. Perfusion imaging, commonly referred to as dynamic susceptibility contrast (DSC), shows increased vascularity in the most malignant portion of the tumor. On patients’ samples, the BBB was assessed by immunofluorescence using anti-Glut1, anti-ZO-1, and anti-Claudin-5 antibodies. Claudin-5 is a membrane protein and component of tight junction strands that serve as a physical barrier to prevent solutes and water from passing freely between epithelial or endothelial cell sheets [27]. 

As expected, in low-grade glioma (WHO grade II astrocytoma and oligodendroglioma), where Gd-enhanced MR indicates no BBB disruption, the ZO-1 immunofluorescence showed a continuous signal lining the Glut-1 positive vessels (Appendix A). Anaplastic astrocytoma and oligodendroglioma (WHO grade III) are likely to represent the ideal specimens to assess the relationship between perivascular tumor invasion and BBB disruption. In these malignant gliomas, only some areas enhanced on Gd-MR, whereas variable portions of tumor do not show any Gd-enhancement, suggesting BBB preservation (Figure 2A,B). As expected, regions with Gd-enhancement showed tumor cells surrounding vessels with disrupted BBB (Figure 2C–E and Appendix A). Conversely, in tumor regions that did not enhance on Gd-MR, in which fluid-attenuated inversion recovery (FLAIR)-MR revealed a hyperintense signal suggesting tumor invasion, the BBB appeared well preserved on immunohistochemistry in spite of the many tumor cells coming in contact with the vascular surface (Figure 2E and Figure 3, and Appendix A). Interestingly, the number of glioma cells that came in close spatial relationship with the vascular endothelium was significantly greater in vessels with preserved BBB, suggesting that perivascular invasion and BBB integrity are related phenomena (Figure 3E). In GBM tumors, the area of surgical resection that enhanced brightly on Gd-MR showed perivascular tumor cells associated with BBB disruption (Figure 4, Appendix A). Again, in GBM regions with hyperintense FLAIR signal that did not enhance on Gd-MR, the BBB of vessels surrounded by tumor cells was not disrupted, as demonstrated by the continuous ZO-1 immunoreaction of Glut-1 positive vessels and by Claudin-5 expression of lectin-positive vessels (Figure 4C,D, Appendix A). 

### 2.3. Relationships of Invading Glioma Cells and Perivascular Astrocytes in GSC Brain Xenografts and Patients’ Tumors 

Perivascular astrocytes with their endfeet are commonly thought to represent an obstacle for the migration of glioma cells around the endothelial wall. Invasion along blood vessels would imply insertion of the tumor cells between the endfeet and the endothelial wall with breach of the BBB [5]. Alternatively, perivascular spreading of glioma cells might occur outside the astrocyte covering without displacement of astrocytes and disruption of BBB. We then examined the relationship between glioma cells, astrocytes, and blood vessels using cell type-specific markers and confocal microscopy in GSC xenografts and human specimens. In GSC xenografts, we found a variety of spatial relationships between perivascular astrocytes, glioma cells, and endothelial cells that included, 1) astrocytic covering maintenance, glioma cells laying outside the astrocyte layer, BBB marker expression maintained by the endothelial cells (Figure 5A, upper panel), 2) displacement of astrocytes, glioma cells coming in direct contact with the endothelium that maintained its BBB marker expression (Figure 3 and Figure 5A, middle panel) glioma cells displacing both astrocytes and endothelial cells with reduced BBB marker expression (Figure 5A, lower panel). Immunohistochemistry of human specimens confirmed the GSC findings, that perivascular glioma cells could either respect the astrocyte covering and BBB integrity (Figure 5B, upper panel), or displace the astrocytic endfeet away from the vasculature coming in direct contact with endothelial cells, which lose their BBB marker expression (Figure 5B, lower panel).

### 2.4. Characterization of Tumor Endothelium in Malignant Glioma by Transmission Electron Microscopy 

Transmission electron microscopy (TEM) was performed in 4 human specimens, that included one grade II astrocytoma, one grade III astrocytoma, and 2 GBM (Cases 2, 4, 20, and 21 in Appendix A). Tumor areas showing various degree of enhancement on Gd-MR were dissected during surgery by MR-assisted neuro-navigation. Blood vessels were analyzed for ultrastructural evidence of permeability routes, including inter-endothelial junctions, basal lamina, and astrocytic endfeet. In grade II astrocytoma, the ultrastructure of vascular endothelium appeared preserved (Figure 6A and Appendix A). In those regions of grade III astrocytoma and GBM that showed a hyperintense FLAIR signal without enhancement on Gd-MR, the tissue ultrastructure appeared less conserved, though the inter-endothelial junctions looked still structured with astrocytic endfeet covering the basal lamina (Figure 6B). The vascular endothelium was characterized by focal points at cells’ inter-junctions structured with the classical “kissing point” and basal lamina wrapped by astrocytic endfeet. In these regions, several tumor cells, which could be recognized because of their high heterogeneity, were found anchored to blood vessels (Figure 6C). The inter-endothelial junctions were maintained, however, processes of tumor cell appeared juxtaposed to blood vessels. Both the basal lamina and endothelial cell appeared enlarged and swollen, however, the endothelial cell junctions were well visible. Where the tumor cells lay juxtaposed to a capillary, both the basal lamina and the endothelial cell appeared thickened and swollen, however, the endothelial cell–cell junctions were still present showing the classical electron density beneath the cell’s membrane (Figure 6C). Regions of grade III astrocytoma and GBM with Gd-enhancement on MR, showed phenomena like erythrocyte extravasation and lack of neuronal tissue. In these regions, the basal lamina appeared thickened, a few axonal fibers were still preserved but the overall tissue morphology appeared highly degenerated. Inter-endothelial junctions with enlarged distensions that may represent sections through trans-endothelial channels, were seen in vessels from Gd-enhancing regions, however, large gaps in the endothelial layer were not seen in GBM vessels.

Interestingly, in the GBM that recurred after radio-chemotherapy, tumor cells with highly heterogeneous morphology arranged themselves to form tubular networks that enclosed erythrocytes within their luminal side, suggesting vascular mimicry (Appendix A). 

To summarize, in malignant gliomas TEM analysis of the FLAIR hyperintense regions without Gd-enhancement on MR showed that the ultrastructure of the brain endothelium, in particular the cell inter-junctions and basal lamina, were preserved even in the condition where tumor cells laid adjacent to blood vessels. 

## 3. Discussion

A novel and unexpected finding of this study is that GBM tumor cells invade the perivascular spaces more extensively and for the longest distances along vessels with preserved BBB. We found that disruption of the BBB results in extravasation of serum-borne molecules, nodular growth of the tumor, and decreased brain invasion. Recently, Watkins and colleagues used GBM xenolines to demonstrate that human glioma cells are able to invade extensively the brain far from the tumor bulk spreading along the perivascular spaces and, more importantly, that in this process glioma cells displace the end feet of astrocytes breaching the BBB [5]. Potentially, these results carry a high translational impact, given that the invading tumor cells would be no more protected from chemotherapy agents, including those drugs that under normal conditions do not cross the BBB. Our findings on brain xenografts of human GBM cells only partly supported the observations by Watkins et al. Specifically, in U87MG xenografts, where rare tumor cells are able to invade the brain away from the main tumor bulk, the BBB is highly disrupted. In brain xenografts of patient-derived GSCs, the occurrence of glioma cells traveling along the perivascular spaces away from the tumor bulk is quite common, however, the event that these cells do breach the BBB is infrequent. A much more common condition is that in which GSCs migrate along vessels with preserved BBB. Surprisingly, those glioma cells that migrated for longest distances, like the contralateral caudate-putamen, were spatially associated with vessels showing preserved BBB. Therefore, the observation that glioma cells are able to displace the endfeet of astrocytes and to breach the BBB can be confirmed only in the U87MG brain xenografts that represents a poor model of human pathology. In the patient-derived GSC model, we observe a variety of interactions between glioma cells and perivascular astrocytes, most of which do not imply disruption of the BBB.

Differences in in vivo models may be influential in explaining such discrepant results. For example, the xenolines used by Watkins et al. were grown in the subcutaneous tissue, an environment that is not protected against serum-borne molecules. Growing in the subcutaneous tissue may induce a tumor phenotype more aggressive towards the BBB. As a support to this concept, we found that the human U87MG cells, which are serum cultured cells, do breach the BBB extensively. 

Our results suggest that BBB breach, exposure to serum-borne molecules, and infiltrative tumor growth are closely related phenomena. In U87MG brain xenografts, which show nodular growth with poor invasive behavior, a massive extravasation of immunoglobulins occurred both in the tumor core and in the periphery. From this, we can infer that the tumor cells are chronically exposed to serum-borne molecules. Perivascular U87MG tumor cells, owing to the preferential exposure to serum-borne molecules, change their metabolism and growth pattern and such changes are maintained even when the tumor cells are removed from their vascular association and secondarily grafted onto the brain of naive recipients [28]. Differentiation conditions, like treatment with serum, reduce the invasive behavior of GSCs [29]. It is known that serum is often used as an inducer of differentiation in in vitro GBM cell models and that the time to serum exposure affects the degree of differentiation, whereby GBM neurospheres exposed to serum for long time show highly reduced infiltrating growth in brain xenografts [30]. 

Glioma heterogeneity is reflected in the variable amount of BBB preservation across WHO grades and also in the very same histotype [31]. Thus, in human specimens of glioma (WHO grade II to IV), we assessed tumor areas with various degrees of BBB disruption, as revealed by the amount of Gd enhancement on MR, that were targeted intraoperatively by navigational systems. Tumor specimens were analyzed using both immunofluorescence and TEM. In anaplastic astrocytoma and oligodedroglioma (WHO grade III), the great majority of the tumor did not enhance on Gd-MR, suggesting a preserved BBB. In these areas, however, immunohistochemistry showed several tumor cells travelling along peri-vascular spaces without BBB breach. The same result was found in those regions of GBM that showed a hyperintense FLAIR signal but did not enhance on Gd-MR, suggesting brain invasion without BBB disruption. Importantly, TEM showed that the BBB ultrastructure, in particular the inter-endothelial junctions and basal lamina, can be preserved even in those vessels that lay adjacent to tumor cells. 

Future studies on the molecular profiling of endothelial cells, pericytes, and astrocytes in primary brain tumor will reveal previously unknown features of the BBB. For example, genetic and transcriptomic analyses of normal brain endothelial cells have shown the activation of WNT–β-catenin and sonic hedgehog (SHH)-dependent signaling within the BBB (for review see Arvanitis et al., 2019 [31]). Performing single-cell RNA sequencing of brain endothelial cells, pericytes, and glia, isolated by specific marker expression, in primary brain tumor and comparing it with normal BBB will reveal more unique properties of the brain neurovascular unit and yield novel therapeutic strategies. Molecular approaches, like GLUT1 targeting and RNA interference to reduce the expression levels of tight junction proteins, can be employed to transiently modulate BBB permeability testing the hypothesis that serum-borne molecules may induce differentiation of glioma cells in vivo.

## 4. Materials and Methods 

### 4.1. Compliance with Ethical Standards

Experiments involving animals were approved by the Ethical Committee of the Università Cattolica del Sacro Cuore (UCSC), Rome (Pr. No. CESA/P/51/2012). This report was drafted according to the ARRIVE guidelines. All patients provided written informed consent to the study according to research proposals approved by the Institutional Ethics Committee of Fondazione Policlinico Gemelli, UCSC (Prot. 4720/17).

### 4.2. Culture of Tumor Cells and Lentiviral Infection 

The U87MG human GBM cell line was purchased from the American Type Culture Collection (Manassas, VA) and cultured in DMEM 4.5 g/L glucose (ThermoFisher Scientific, Waltham, MA, USA) supplemented with 10% Fetal Bovine Serum (ThermoFisher Scientific). The patient-derived GSC1 and GSC275 cell lines were cultured under serum-free conditions [6,7]. Cells were grown at 37 °C in a humidified atmosphere of 5% CO_2_–95% air. Cells were regularly controlled to exclude mycoplasma contamination (Mycoalert Detection Kit, Lonza, Basel, Switzerland). Lentiviral transduction of green fluorescent protein (GFP) was performed as described [32].

### 4.3. Intracranial Xenografts of GFP Expressing U87MG, GSC1, and GSC275 Cells

Immunosuppressed athymic rats (male, 250–280 g; Charles River, Milan, Italy) were used for brain xenografts [26] (Appendix A). 

### 4.4. Fluorescence Microscopy and Immunofluorescence of Brain Tumor Xenografts 

The brains were serially sectioned (40 μm thickness) on the coronal plane, blocked in PB with 10% BSA, 0.3% Triton X-100 for 45 min, and incubated overnight at 4 °C with primary antibodies (Appendix A) [33,34]. Monoclonal antibodies used were as follows, mouse anti-Glucose Transporter GLUT1 antibody (1:100; Abcam, Cambridge, UK), mouse anti-Rat Blood-Brain Barrier (Clone SMI-71; 1:500; Biolegend, San Diego, CA, USA), mouse anti-Claudin-5 (1:100; Thermo Fisher Scientific, Waltham, MA, USA). Polyclonal antibodies used were as follows, rabbit anti-Glucose Transporter GLUT1 antibody (1:200; NovusBio, Centennial, CO, USA), rabbit anti-ZO-1 (1:100; Thermo Fisher Scientific, Waltham, MA, USA), goat anti-GFAP (1:1000; Thermo Fisher Scientific, Waltham, MA, USA), rabbit anti-GFAP (1:1000; Dako Italia, Milan, Italy). For detecting brain microvessels, sections were incubated with Lectin from *Lycopersicon esculentum* (tomato) biotin conjugate (1:500; Sigma-Aldrich, St. Louis, MO, USA) together with primary antibodies (Appendix A). To detect vascular permeability in brain xenografts, sections were incubated with Alexa Fluor 555 donkey anti-rat IgG (1:100; Abcam, Cambridge, UK) together with other primary antibodies (Appendix A). Immunofluorescence was observed with a laser confocal microscope (Leica SP5 or Olympus FV1200). Image analysis was performed with Leica Application Suite X software and ImageJ software (NIH).

### 4.5. Clinical Material

Tumor specimens were obtained during craniotomy surgery. All patients provided written informed consent to the study according to research proposals approved by the Institutional Ethics Committee of Fondazione Policlinico Gemelli, UCSC (Prot. 4720/17). The study was conducted in accordance with the principles set forth in the World Medical Association Declaration of Helsinki and later amendments. Tumor regions showing hyper-intense FLAIR signal and various degrees of enhancement on Gd-MR were located during surgery using a neuro-navigational system (SthealthStation S7, Medtronic, Minneapolis, MN, USA). Specimens were fixed in 4.5% formalin for 48 h at 4 °C, post-fixed in 30% sucrose, and sectioned (40–50 μm) by a cryostat. Slices were incubated overnight at 4 °C in PB with 0.3% Triton X-100 and 0.1% NDS with Lectin from Lycopersicon esculentum (tomato) biotin conjugate (1:500; Sigma-Aldrich, St. Louis, MO, USA) together with primary antibodies. Sections were incubated overnight at 4 °C in PB with 0.3% Triton X-100 and 0.1% NDS with Lectin from *Lycopersicon esculentum* (tomato) biotin conjugate (1:500; Sigma-Aldrich, St. Louis, MO, USA) in combination with other antibodies. Monoclonal antibodies used were as follows, mouse Anti-IDH1 (R132H; clone HMab-1, 1:50, Sigma Aldrich, St. Louis, MO, USA); rat anti-Collapsin Response-Mediated Protein 5 (CRMP5, 1:50, Millipore, Billerica, MA, USA); mouse anti-Claudin-5 (1:100; Thermo Fisher Scientific, Waltham, MA, USA). Polyclonal antibodies used were as follows, rabbit anti-Glucose Transporter GLUT1 antibody (1:200; NovusBio, Centennial, CO, USA); rabbit anti-ZO-1 (1:100; Thermo Fisher Scientific, Waltham, MA, USA); goat anti-GFAP (1:1000; Thermo Fisher Scientific, Waltham, MA, USA).

### 4.6. Antigen Retrieval and Auto-Fluorescence Removal in Brain Tumor Xenografts and Human Specimens

To unmask CRMP5, ZO-1, and Claudin5 antigens, and to reduce the masking effect of formalin fixation specific procedures were needed before immunostaining (Appendix A). 

### 4.7. Transmission Electron Microscopy (TEM)

Specimens were fixed with 2% (*w*/*v*) PFA, 2.5% Glutaraldehyde (TAAB) in 0.1 M Phosphate Buffer overnight at 4 °C. Samples were carefully washed in 0.1 M Na Cacodylate buffer pH 7.1 (Science Service), postfixed in 1% OsO4 (TAAB) supplemented with 1.5% Potassium Ferrocyanide for 2 h on ice (Science Service), and counterstained with 1% aqueous Uranyl Acetate ON at 4 °C. Samples were subsequently dehydrated in Ethanol and infiltrated with propylene oxide/Durcupan (Science Service) (1:1) followed by pure resin embedding and polymerization for 72 h at 60 °C. Ultrathin sections were cut (Ultracut S, Leica, Mannheim, Germany) and observed with a Transmission Electron Microscope (TEM) Jeol 1010 equipped with a MSC 791 CCD camera (Gatan).

### 4.8. Statistical Analysis

Comparison between density of vascular structures and distance from the tumor margin was performed using Student’s *t*-test. Correlation between GFP expressing tumor cells and SMI-71 expressing vessels was evaluated using regression analysis and the Spearman correlation test. All *P*-values are based on 2-tailed tests and differences were considered significant when *p* < 0.5. StatView v5.0 was used (SAS Institute, Cary, NC, USA).

## 5. Conclusions

Previous findings showing that single glioma cells disrupt the BBB far away from the tumor bulk are not confirmed in brain xenografts of patient-derived GSCs and in surgical specimens of malignant glioma. The present study suggests a model of malignant glioma where phenomena like BBB disruption and exposure to serum-borne molecules elicit a nodular growth of the tumor bulk. Conversely, BBB preservation and perivascular growth characterize brain invasion outside the tumor bulk.

## Figures and Tables

**Figure 1 cancers-12-00018-f001:**
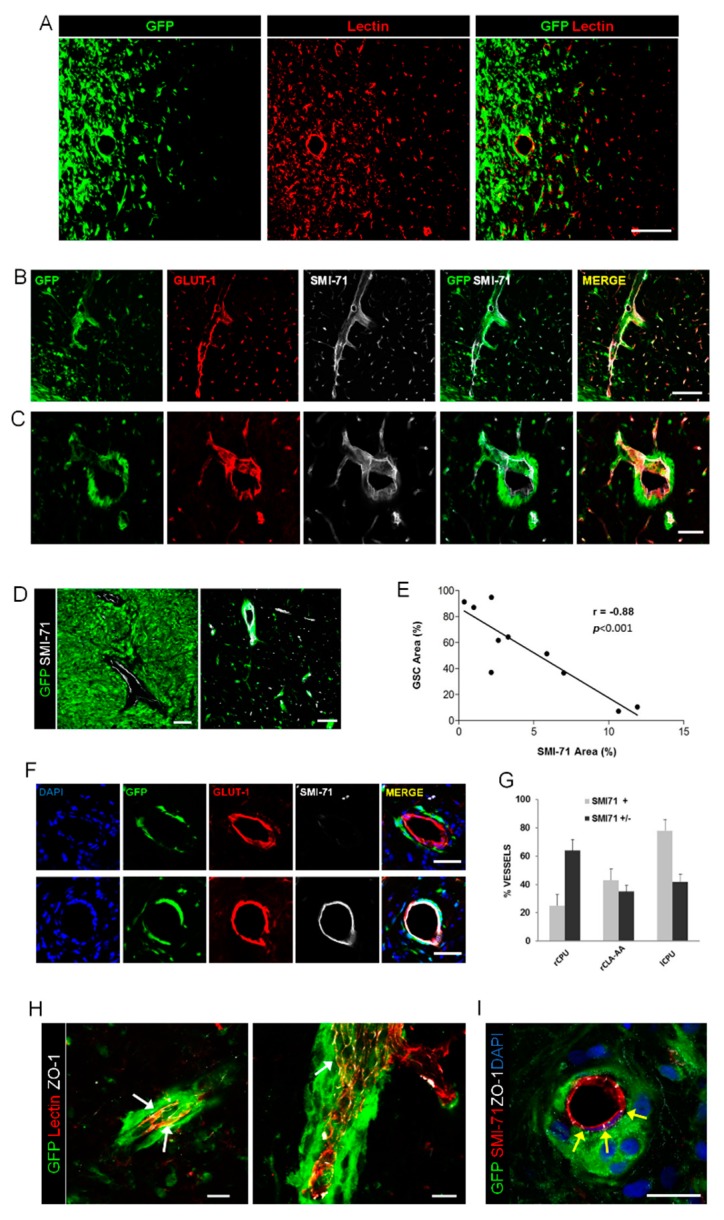
Brain xenografts of GSC1 cells. (**A**) Fluorescence microscopy of the brain–tumor interface showing invading glioma stem-like cells (GSCs) and remarkable angiogenesis. Scale bar, 150 μm. (**B**,**C**) GSCs extensively spread around vessels that maintained their SMI-71 expression. Scale bar in B, 150 μm. Scale bar in C, 50 μm. (**D**) In the core of GCS xenografts (left panel), the vessels showed a consistent reduction of SMI-71 immunostaining, whereas in the infiltrated brain away from the tumor bulk (right panel) the expression of SMI-71 by the blood vessels was preserved. Scale bars, 100 μm. (**E**) Diagram showing the relationship between tumor cells density and SMI-71 expression by endothelial cells, as assessed by automated image analysis (each point represents an average of 7–12 areas; r, Pearson correlation coefficient). (**F**) Representative vessels showing peri-vascular spreading of GSCs (green) and various degrees of BBB disruption on SMI-71 immunostaining (white). Scale bars, 50 μm. (**G**) Bar diagram showing that the percent of vessels with perivascular tumor cells and preserved SMI-71 expression (SMI71+) was significantly higher in the caudate-putamen contralateral to the xenograft (lCPU) compared to the homolateral caudate-putamen (rCPU) and claustrum-amygdaloid area (rCLA-AA). Bars represent mean+sem; (**H**,**I**) Immunofluorescence microscopy showing that the expression of ZO-1 (white arrows) is well preserved even in vessels surrounded by multilayered GSCs. Scale bars, 25 μm.

**Figure 2 cancers-12-00018-f002:**
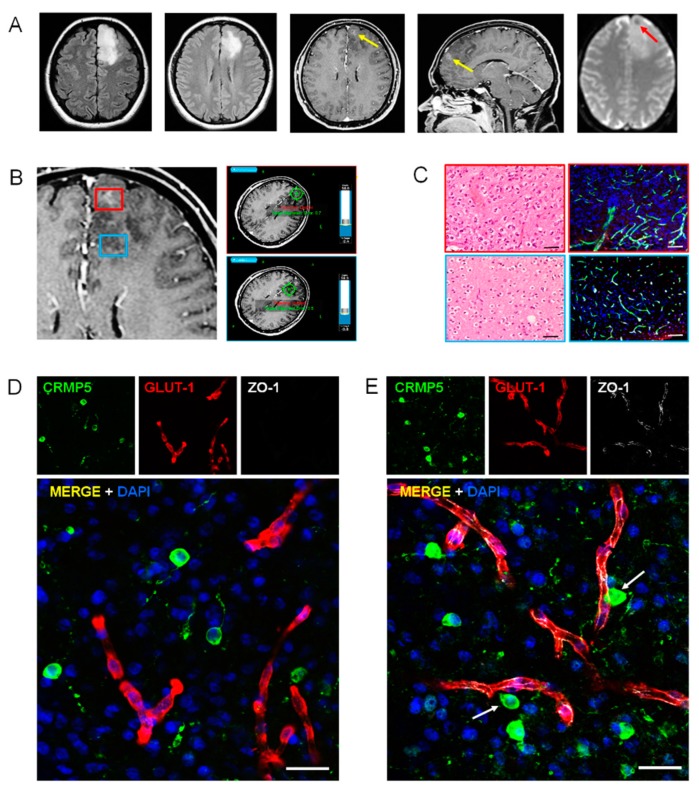
Perivascular invasion and disruption of blood–brain barrier (BBB) in anaplastic glioma. (**A**), FLAIR T2–weighted, Gd-enhanced T1-weighted, and DSC perfusion magnetic resonance (MR) images showing a left frontal tumor that shows a hyperintense signal in the fluid-attenuated inversion recovery (FLAIR) sequences. The small area of Gd enhancement (yellow arrows) and low T2 signal (red arrow) suggests focal breach of BBB with increased vascularity. (**B**) Neuro-navigational procedure for selecting tumor regions with or without Gd enhancement (red and blue insets, respectively). (**C**) Histology and immunohistochemistry of Gd-enhancing (red inset) and non-enhancing (blue inset) areas of the tumor (anaplastic oligodendroglioma). Left panels, H&H staining. Right panels, double anti-CD31 (green) and anti-IDH1 (red) immunofluorescence. Scale bars, 100 μm. (**D**,**E**) Immunofluorescence with the anti-CRMP5 antibody for tumor cells (green) and with anti-Glut-1 (red) and anti-ZO-1 (white) antibodies for BBB in a tumor region showing Gd-enhancement (**D**) and in a region without Gd-enhancement (**E**). The arrows point out tumor cells adjacent to vessels with normal ZO-1 expression. Scale bars, 25 μm.

**Figure 3 cancers-12-00018-f003:**
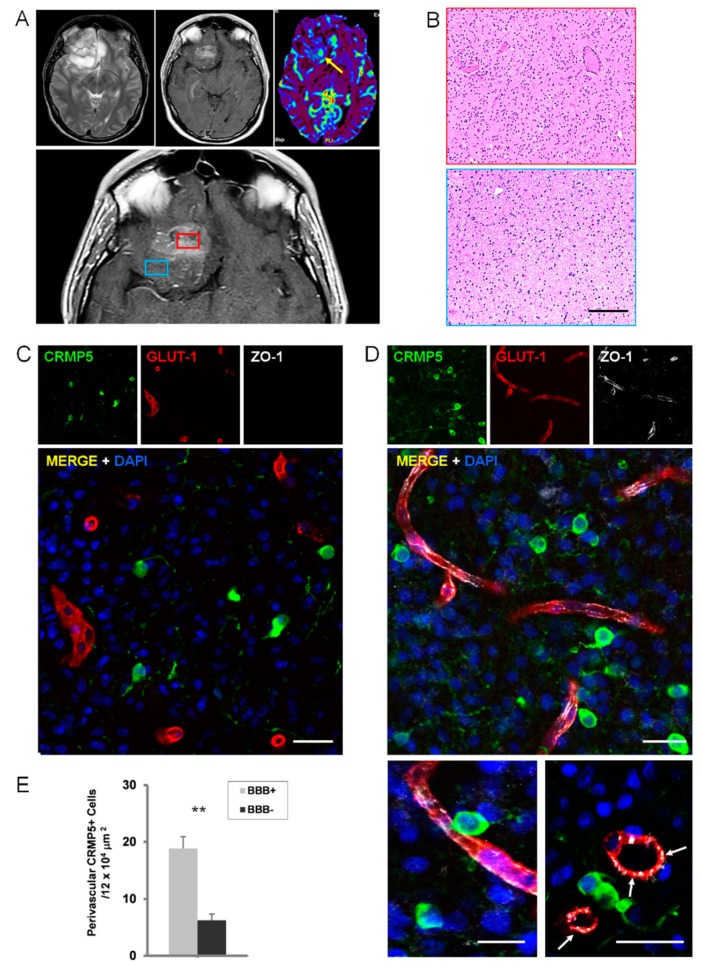
Perivascular invasion and disruption of BBB in anaplastic glioma. (**A**), FLAIR T2–weighted (top left), Gd enhanced T1-weighted (top center), and DSC perfusion (top right) MR images showing a left intrinsic fronto-basal tumor that shows an inhomogeneous hyperintense signal in the FLAIR sequence and an area of Gd enhancement with increased vascularity (yellow arrow). Neuro-navigation selecting tumor regions with or without Gd enhancement (red and blue insets, respectively; lower panel). (**B**), Histology of Gd-enhancing (red inset) and non-enhancing (blue inset) areas of the tumor (anaplastic oligodendroglioma). H&H staining. Scale bar, 100 μm. (**C**,**D**), Immunofluorescence with the anti-CRMP5 antibody for tumor cells (green) and with anti-Glut-1 (*red*) and anti-ZO-1 (*white*) antibodies for BBB in a tumor region showing Gd-enhancement (**C**) and in a region without Gd-enhancement (**D**). The arrows point out tumor cells adjacent to vessels showing ZO-1 expression. Scale bars, 25 μm. (**E**), Graph showing the density of perivascular CRMP5 tumor cells in regions of anaplastic gliomas (*n*, 4) with preserved (BBB+) or disrupted BBB (BBB-), as assessed by ZO-1 immunohistochemistry. **, *p* < 0.0001 (Student-*t* test).

**Figure 4 cancers-12-00018-f004:**
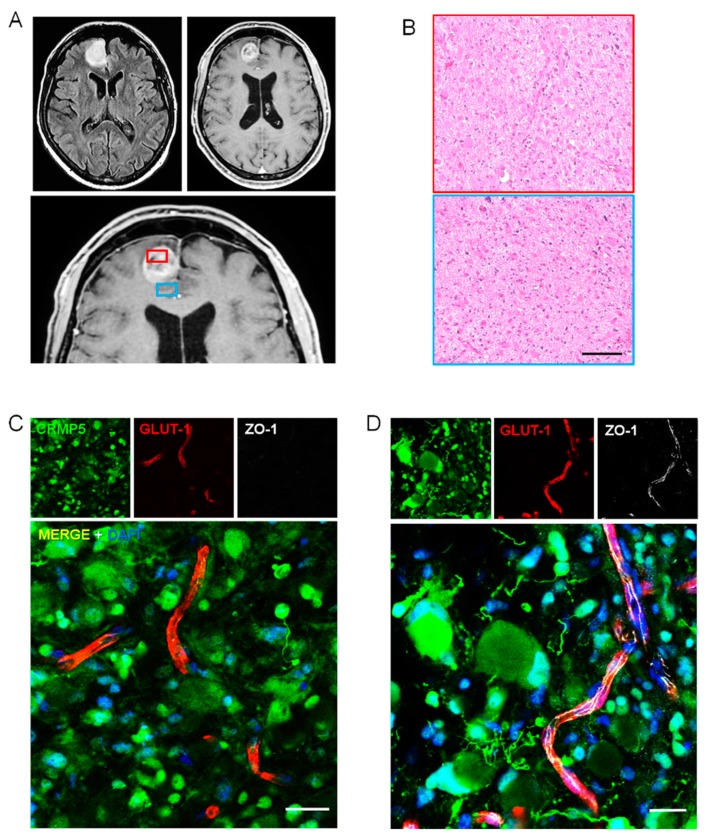
Perivascular invasion and disruption of BBB in glioblastoma. (**A**), FLAIR T2–weighted (top left panel) and Gd enhanced T1-weighted (top right and lower panels) MR images showing a small left frontal tumor that shows a hyperintense signal in the FLAIR sequence and ring enhancement on Gd-MR. Neuro-navigation for selecting tumor regions with or without Gd enhancement (red and blue insets, respectively). (**B**) Histology of Gd-enhancing (red inset) and non-enhancing (blue inset) areas of the tumor. H&H staining. Scale bar, 100 μm. (**C**,**D**) Immunofluorescence with the anti-CRMP5 antibody for tumor cells (green) and with anti-Glut-1 (red) and anti-ZO-1 (white) antibodies for BBB in a tumor region showing Gd-enhancement (**C**) and in a region without Gd-enhancement (**D**). Note the strong ZO-1 expression by a vessel surrounded by tumor cells with gemistocytic appearance. Scale bars, 25 μm.

**Figure 5 cancers-12-00018-f005:**
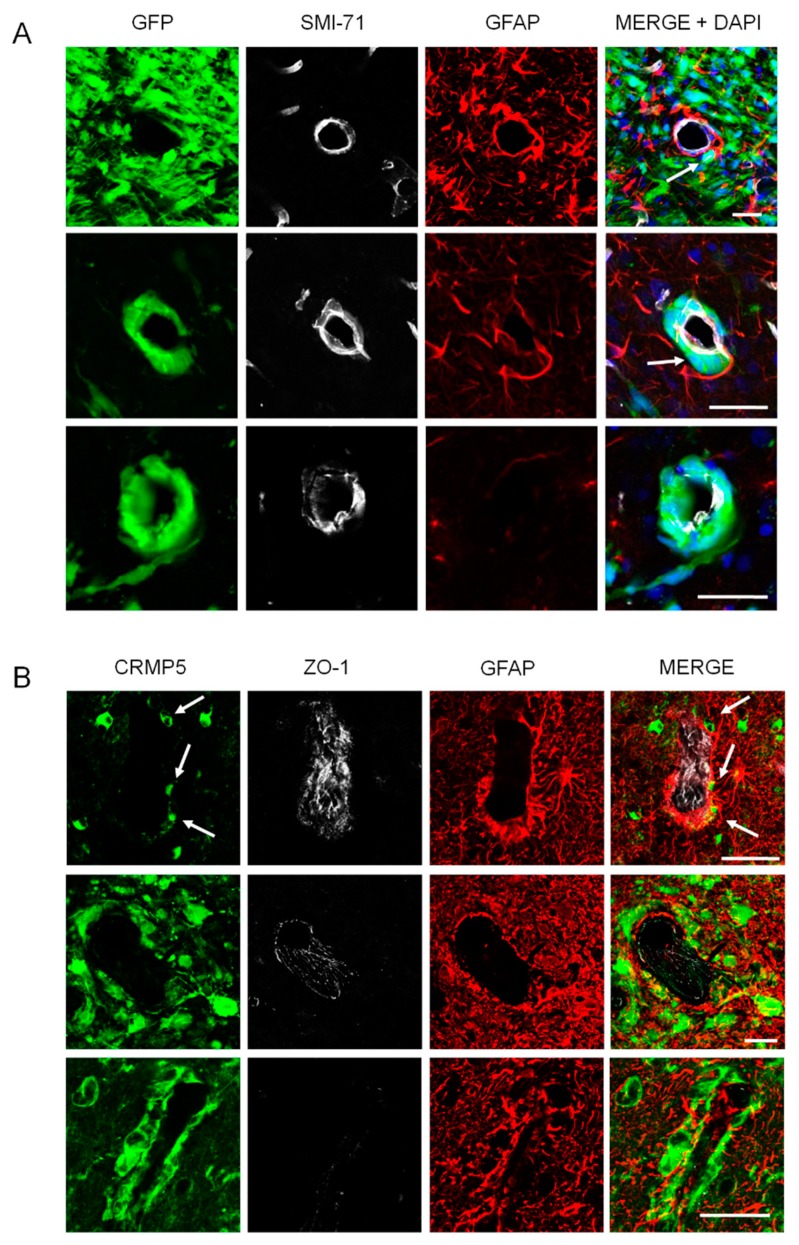
Relationships of invading glioma cells and perivascular astrocytes. (**A**), In GSC brain xenografts, perivascular tumor cells (green) either lay outside the astrocyte covering without astrocyte displacement and BBB disruption (top panel, arrow), or displace the astrocytes coming in direct contact with the endothelium that maintains its SMI-71 expression (middle panel, arrow), or displace completely the astrocytes with partial loss of SMI-71 expression (lower panel). Scale bars, 50 μm. (**B**) In surgical specimens of GBM, tumor cells (green) were found outside the astrocyte covering with only minor displacement of astrocytes and preservation of ZO-1 expression (top and middle panels, arrows) or displace completely the astrocytes and disrupt the vessel wall, which loses ZO-1 expression (lower panel). Scale bars, 50 μm.

**Figure 6 cancers-12-00018-f006:**
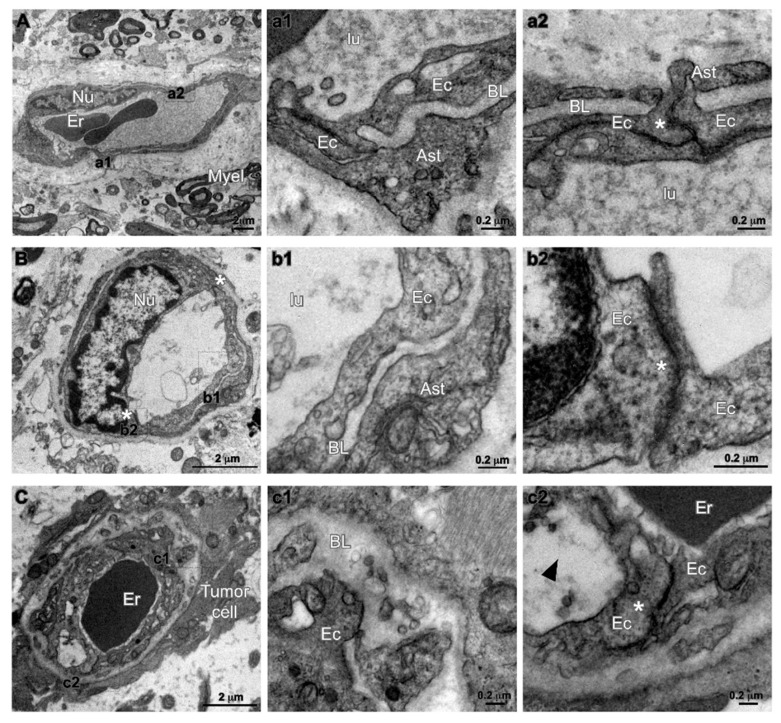
Transmission electron microscopy (TEM) analysis of human glioma specimens. (**A**) Representative image of well-structured capillary (*Cap*)-enclosing erythrocytes (*Er*) in a low grade astrocytoma (WHO grade II). Tissue and myelinated axonal fibers are still preserved (*Myel*). *Nu*, nucleus of endothelial cell. High magnification insets (a1,a2), both indicate “kissing” points of endothelial cell (*Ec*) tight junctions (*asterisk*), and astrocytic endfeet (*Ast*) covering the basal lamina (*BL*). (**B**) Panel showing a capillary (*Cap*) in a region of anaplastic astrocytoma (WHO grade III) without Gd-enhancement. The inter-endothelial junctions are maintained at both sides (*asterisks*) of endothelial cell nucleus (*Nu*). Insets (b1,b2) show respectively the luminal part of the vessel (*lu*) coated by the endothelial wall (*Ec*), the basal lamina (*BL*) and the astrocytic endfeet (*Ast*). (**C**) panel representative of region of anaplastic astrocytoma (WHO grade III) without Gd-enhancement. Low magnification micrograph showing processes of tumor cell juxtaposed to a blood vessel (*Cap*) that enclose an erythrocyte (*Er*). In high magnification insets (c1,c2), both the basal lamina (*BL*) and the endothelial cell (*arrowhead*) appear enlarged and swollen, although endothelial cell junctions are still present (*asterisk* in inset c2).

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
