# Peer review of "Brain Invasion along Perivascular Spaces by Glioma Cells: Relationship with Blood–Brain Barrier"

_cancers, 2019, doi:10.3390/cancers12010018_

Round 1
Reviewer 1 Report
In the current manuscript Pacioni et al. describe that during the invasion process of brain tumors the blood-brain barrier is retained at the vessels sides hijacked as migration tracks for the invading brain tumor cells. This is an interesting and very well done study with high translational potential. However, a few points remain to be answered:
In the presence of a waste access of data on glioma and the blood-brain barrier what is new in this study? The authors mix data derived of low grad and secondary glioblastoma (with IDH1/2 mutation) but present a conclusion for glioma in general. However, the term glioma is commonly used for malignant brain tumors grade IV or primary glioblastoma, a detrimental and incurable disease. However, these tumors have not been studied and therefore the conclusion is misleading as most functional data has been compiled using low grade glioma. The authors should revisit the manuscript and in particular change the title to make this point clear. How frequent has the marker CRMP5 been used for glioma cells so far or how established is it? The WHO grade system II-IV used in this manuscript is outdated. Which tumor subgroups according to the actual molecular classification correspond to the tumors used here? The font size should be corrected as it varies throughout the text.Author Response
Reviewer 1
In the current manuscript Pacioni et al. describe that during the invasion process of brain tumors the blood-brain barrier is retained at the vessels sides hijacked as migration tracks for the invading brain tumor cells. This is an interesting and very well done study with high translational potential. However, a few points remain to be answered:
In the presence of a waste access of data on glioma and the blood-brain barrier what is new in this study? The authors mix data derived of low grad and secondary glioblastoma (with IDH1/2 mutation) but present a conclusion for glioma in general. However, the term glioma is commonly used for malignant brain tumors grade IV or primary glioblastoma, a detrimental and incurable disease. However, these tumors have not been studied and therefore the conclusion is misleading as most functional data has been compiled using low grade glioma. The authors should revisit the manuscript and in particular change the title to make this point clear.
As shown in Supplementary Table S2 (Clinical and Pathological Features of Patients’ Tumors), we studied 3 cases of low-grade glioma, 4 cases of anaplastic glioma with IDH1/2 mutation, 1 case of secondary glioblastoma with IDH1/2 mutation, and 13 cases of primary glioblastoma IDH1/2 wt. Therefore, our conclusions are based above all on malignant tumors grade IV or primary glioblastoma.
How frequent has the marker CRMP5 been used for glioma cells so far or how established is it?
In this study, we used the antibody against Collapsin Response Mediator Protein 5 (CRMP5). This antibody has recently been proposed as a selective tumor marker for cancer cells and glioma cells (PMID:26122847; 23298946).
In a previous work from our group, we validated this antibody on cultured human glioma cells with mutant IDH1, where the anti-CRMP5 antibody co-stained the IDH1/2 mutant cells (PMID:30414187).
In the present study, we further validated the anti-CRMP5 antibody using surgical specimens of low-grade glioma (WHO grade II astrocytoma and oligodendroglioma) with mutant IDH1/2 and showed a co-staining of the tumor cells with anti-IDH1 and anti-CRMP5 antibodies (Supplementary Fig. S4 and Supplementary Table S2).
In addition, among the glioblastoma cases included in our study, is there a giant cell glioblastoma (Fig. 4), in which the giant cells can be easily recognized because of their morphology and in which these cells strongly stained with the anti-CRMP5 antibody.
In the revised version of our manuscript, we added one more reference concerning the use of this antibody for the selective staining of cancer cells.
Brot, S.: Malleval, C.; Benetollo, C.; Auger, C.; Meyronet, D.; Rogemond, V.; Honnorat, J.; Moradi-Améli, M. Identification of a new CRMP5 isoform present in the nucleus of cancer cells and enhancing their proliferation. Exp. Cell Res. 2013, 319, 588-599. PMID:23298946
The WHO grade system II-IV used in this manuscript is outdated.
In our manuscript, we used the 2016 WHO grade system II-IV that is the latest version of the WHO classification of brain tumors (Louis DN, Ohgaki H, Cavenee WK, Ellison DW, Figarella-Branger D, Perry A, Reifenberger G, von Deimling A. WHO Classification of Tumours of the Central Nervous System, revised 4th ed; International Agency for Research on Cancer: Lyon, France, 2016; pp. 10-77.). According to this grading system, other than the histological picture, the diagnosis of glioblastoma implies the distinction between primary glioblastoma IDH1/2 wt and secondary glioblastoma IDH1/2 mutant (Supplementary Table S2). For establishing the diagnosis of oligodendroglioma (WHO grade II and III), other than IDH1/2 mutation, the 1p19p co-deletion was assessed (Supplementary Table S2).
Which tumor subgroups according to the actual molecular classification correspond to the tumors used here?
We thank this Reviewer for his comment that raises a relevant issue, i.e., that the tumor subtype may affect the ability of the glioma cells to spread for long distances along the perivascular spaces. This issue may be difficult to assess using surgical samples, in which the extend of resection includes only a small portion of the hyperintense T2-FLAIR region that does not enhance on Gd infusion. However, the relationship between tumor subtype and degree of perivascular invasion can be investigated using the brain xenograft model. In the previous version of our manuscript, we used brain grafts of the patient-derived GSC1 cell line. This cell line had been established from a glioblastoma of the proneural subtype and was previously characterized molecularly as a Glioma Stem full (GSf) cell line, a genotype that closely resemble the proneural one (PMID:28248456).
In the revised manuscript, we added an experiment that used brain xenografts of GSC275 cells, a cell line that had been raised from a glioblastoma of the mesenchymal subtype and that was previously characterized molecularly as a Glioma Stem restricted (GSr) cell line, resembling the mesenchymal subtype (PMID:28248456). In GSC275 brain xenografts, we found perivascular tumor cells spreading at distant sites from the bulk of the tumor. Importantly, even in the mesenchymal xenografts, the BBB of such vessels was not disrupted. In the revised manuscript, the new Supplementary Figure S4 has been incorporated showing perivascular invasion and BBB integrity in the brain xenografts of 275 cells (GSr subtype or mesenchymal-like).
The font size should be corrected as it varies throughout the text.
The font size has been corrected throughout the text.

Reviewer 2 Report
Pacioni et al. suggested that BBB integrity was preserved in the majority of vessels located outside the tumor bulk using brain xenografts of patient-derived GSCs . This article is very interesting and covers the important issue. However, it is necessary to present clearer description for the publication of "Cancers" in addition to immunostaing and EM . 1. They used brain xenografts of patient-derived glioma stem-like cells (GSCs) expressing GFP and suggested BBB integrity was preserved in the majority of vessels located outside the tumor bulk in spite of massive perivascular invasion. However, it is confusing since title of this paper suggest "Brain invasion along perivascular spaces by glioma cells and disruption of blood brain barrier". Hence, it is required to describe clearer title and abstract. 2. please check line 27, 77 and 284-291 (text bold and different text size) 3. In line 344 (Compliance with Ethical Standards), please describe IRB number for utilization of patient-derived GSCs. 4. In line 356, please describe the method in detail or reference. 5. In discussion, it is interesting if to discuss what is the difference of glioblastoma grade (I-IV) with the respect of BBB integrity or disruption. (Is there any evidence for example other different molecules?) 6. It is important and interesting to examine the effect of important gene on BBB integrity via knock-down or overexpression etc.Author Response
Reviewer 2
Pacioni et al. suggested that BBB integrity was preserved in the majority of vessels located outside the tumor bulk using brain xenografts of patient-derived GSCs . This article is very interesting and covers the important issue. However, it is necessary to present clearer description for the publication of "Cancers" in addition to immunostaing and EM .
1. They used brain xenografts of patient-derived glioma stem-like cells (GSCs) expressing GFP and suggested BBB integrity was preserved in the majority of vessels located outside the tumor bulk in spite of massive perivascular invasion. However, it is confusing since title of this paper suggest "Brain invasion along perivascular spaces by glioma cells and disruption of blood brain barrier". Hence, it is required to describe clearer title and abstract.
We agree with this Reviewer that the title of the paper may be misleading. The title was changed as “Brain invasion along perivascular spaces by glioma cells. Relationship with blood brain barrier”.
2. please check line 27, 77 and 284-291 (text bold and different text size)
We checked line 27, 77 and 284-291 and changed text bold and text size.
3. In line 344 (Compliance with Ethical Standards), please describe IRB number for utilization of patient-derived GSCs.
Approval number for utilization of patient-derived GSCs (Ethics Committee of Fondazione Policlinico Gemelli, Prot. 4720/17) has been added in line 344.
4. In line 356, please describe the method in detail or reference.
Reference 24 has been added describing the technique for brain xenografting in rats.
5. In discussion, it is interesting if to discuss what is the difference of glioblastoma grade (I-IV) with the respect of BBB integrity or disruption. (Is there any evidence for example other different molecules?)
Differences among glioma of various grade (I-IV) with respect of BBB integrity, as assessed by SMI71, Glut1, ZO-1, and Claudin immunohistochemistry, and IgG extra-vasation are shown in Supplementary Table S3.
6. It is important and interesting to examine the effect of important gene on BBB integrity via knock-down or overexpression etc.
We fully agree with this Reviewer that the molecular profiling of the BBB cell actors (brain endothelial cells, pericytes, and astrocytes) in primary brain tumour will reveal previously unknown features. For example, genetic and transcriptomic studies have confirmed the activation of signalling pathways, like WNT–β-catenin and sonic hedgehog (SHH)-dependent signalling in brain endothelial cells within the BBB (for review see PMID:31601988). Recent single-cell RNA sequencing of normal brain endothelial cells, pericytes and glia, isolated by specific marker expression, has provided distinct cell-specific transcriptional profile. Performing single-cell analysis of the BBB in primary brain tumour and comparing it with normal BBB will reveal more unique properties of the brain neurovascular unit and yield novel therapeutic strategies. Other molecular approaches to transiently modulate BBB permeability may include GLUT1 targeting and RNA interference to reduce the expression levels of tight junction proteins. A statement on future studies aimed at examining the effect of important gene on BBB integrity (via knock-down or overexpression) has been included in the Discussion.
Arvanitis, C.D.; Ferraro, G.B.; Jain, R.K. The blood-brain barrier and blood-tumour barrier in brain tumours and metastases. Nat. Rev. Cancer. 2019 (Epub ahead of print); DOI: 10.1038/s41568-019-0205-x. PMID: 31601988

Round 2
Reviewer 2 Report
This revised form is now acceptable for the publication of "Cancers".